# Weakly Supervised Violence Detection in Surveillance Video

**DOI:** 10.3390/s22124502

**Published:** 2022-06-14

**Authors:** David Choqueluque-Roman, Guillermo Camara-Chavez

**Affiliations:** 1Department of Computer Science, Universidad Católica San Pablo, Arequipa 04001, Peru; 2Department of Computer Science, Federal University of Ouro Preto, Ouro Preto 35400-000, Brazil; guillermo@ufop.edu.br

**Keywords:** video surveillance, violence detection, weakly supervised, spatiotemporal violence detection, dynamic image

## Abstract

Automatic violence detection in video surveillance is essential for social and personal security. Monitoring the large number of surveillance cameras used in public and private areas is challenging for human operators. The manual nature of this task significantly increases the possibility of ignoring important events due to human limitations when paying attention to multiple targets at a time. Researchers have proposed several methods to detect violent events automatically to overcome this problem. So far, most previous studies have focused only on classifying short clips without performing spatial localization. In this work, we tackle this problem by proposing a weakly supervised method to detect spatially and temporarily violent actions in surveillance videos using only video-level labels. The proposed method follows a Fast-RCNN style architecture, that has been temporally extended. First, we generate spatiotemporal proposals (action tubes) leveraging pre-trained person detectors, motion appearance (dynamic images), and tracking algorithms. Then, given an input video and the action proposals, we extract spatiotemporal features using deep neural networks. Finally, a classifier based on multiple-instance learning is trained to label each action tube as violent or non-violent. We obtain similar results to the state of the art in three public databases Hockey Fight, RLVSD, and RWF-2000, achieving an accuracy of 97.3%, 92.88%, 88.7%, respectively.

## 1. Introduction

Video surveillance is a valuable tool for monitoring human behaviors such as violence. Many video surveillance cameras are deployed in public and private places such as hospitals, schools, and prisons, to detect and prevent violent behaviors. A human operator can only monitor a few surveillance cameras efficiently, but is frequently burdened with numerous cameras, resulting in errors and missed detections. Thus, an intelligent method for detecting violent actions is necessary to overcome this problem.

Violence detection is a specific task of action recognition. It detects physical violence, such as fights, robberies, arrests, and protests, between two or more people in a video. It is related to other areas of computer vision, such as action recognition [1], object detection [2], and anomaly detection [3]. At first sight, detecting violent actions seems like an easy task. However, this is a challenging task because of the nature of video surveillance data. Usually, video surveillance data contain artifacts such as variable illumination, complex backgrounds, camera movement, and distance between the actors and the camera. These artifacts make the detection of violent actions a problematic task.

In previous years, the search for methods of violence detection was mainly promoted by the MediaEval [4] benchmark between years 2011 and 2014. The violence scene detection task of MediaEval was oriented to detect violent scenes in movies. The definitions of “violence” adopted in MediaEval were a scene that “one would not let an eight years old child see in a movie because they contain physical violence”, and “physical violence or accidents resulting in human injuries or pain”. Both definitions contain a high semantic level that makes their translation difficult for computer vision. We consider a violent action as a physical interaction between people resulting in human injuries or pain in this work.

So far, more research has been conducted on action recognition, while the number of investigations on violence detection has been comparatively less studied. In recent years, violence detection methods have used different techniques such as handcrafted methods and deep convolutional networks. Handcrafted methods design specific descriptors to represent video spatiotemporal features, and they typically use Bag-of-visual-words (BoVW) models. Forinstance, Moreira et al. [5] adapted a detector in a BoVW framework to detect violent actions. Cai et al. [6] have studied techniques such as Dense Trajectories, MPEG flow, Fisher Vectors, and Sparse Codification using BoVW. Zhou et al. [7] detect violent frame sequences into dense environments using descriptors extracted from Optical Flow. Handcrafted feature methods do not require a larger dataset to achieve good results. However, they have generalization problems because of their specific design.

In recent years, deep-learning models have achieved higher accuracy and better results in various computer vision tasks, including violence detection. Xia et al. [8] propose a two-stream CNN combined with an SVM classifier [9] to detect violent actions. Malveira et al. [10] train a multi-stream CNN over specific violent concepts such as fights, shots, and explosions, to decompose violence. Song et al. [11] propose a new pre-processing keyframe-based method and a 3D-CNN model to detect violence more efficiently. Li et al. [12] detect violence between three or more people training two CNN’s using RGB frames and Optical Flow. Li et al. [13] modify a 3D-CNN leveraging dense layers [14] to encourage feature reusing and channel interaction. The main advantage of these methods is the higher generalization capability of the CNN’s. However, they usually require a larger dataset during training [15].

Although the abovementioned approaches are appealing and achieve good performance, most aim to classify a short clip as violence or non-violence (localizing violence in time). However, for real-world applications, the spatial location of the violence is important to identify and track the action. A possible reason for this research gap might be related to the fact that there is no available dataset with spatial annotations (bounding boxes) enclosing the violent action at each frame. Existing datasets such as the Hockey Fight [16], the RWF-2000 [17] and the CCTV-Fights datasets [18] provide only temporal annotations, making it difficult to train traditional fully supervised action detectors [19,20,21,22]. We propose a method to detect violence spatially and temporally for real-world applications such as video surveillance to overcome this drawback. The proposed method relies on a weakly supervised approach based on Multiple Instance Learning (MIL) [23] to avoid the exorbitant cost of annotating current violence datasets with spatiotemporal labels. Similarly to Arnab et al. [24], we propose to learn a spatiotemporal violence detector by treating violent and non-violent videos as bags and action tubes as instances in a bag.

The main novelties and contributions of this work are summarized below.

In contrast to most previous studies, our method can spatially and temporally localize violent actions. We follow a weakly supervised approach to train the spatiotemporal detector using only video-level labels.We propose a method based on deep neuronal networks and a weakly supervised instance-level classifier. We follow a Fast-RCNN [25]-style architecture detector extended temporally with two streams.We adopt a two-stream approach using a 3D-CNN to extract spatiotemporal features of a video clip and a 2D-CNN to extract spatial features of a keyframe (i.e., dynamic image of the clip), which are trained end-to-end.We leverage dynamic images [26] to represent violent motion information in a few images, avoiding the high computational cost of optical flow.Our approach leverages pretrained person detectors and an online linking algorithm to build action tubes that could contain a violent action. An action tube is a sequence of bounding boxes enclosing an action. Then, we train the proposed model to classify each tube as violent or non-violent. This is the first time action tubes are used for violence detection, to the best of our knowledge.A non-motion suppression strategy is proposed to discard further background motion, which uses dynamic images to extract salient regions with motion in a video sequence to improve the tube extraction method.We conducted ablation studies on the two-stream model to examine the effect of the fusion strategy. Moreover, we have compared different violent motion representations, such as optical flow and dynamic images.

The rest of the paper is arranged as follows: in Section 2, we present a brief review of recent related works. In Section 3, we describe our proposed method. Then, in Section 4 we show experimental results to validate our proposal. Finally, the conclusion and future works are discussed in Section 5.

## 2. Related Works

According to the literature, several violence-detection methods mainly use visual information to extract useful features [27]. Traditional hand-crafted-based methods extract visual features using a manually predefined algorithm. Usually, they are designed to overcome specific problems such as occlusions, scale variation, and variable illumination. Alternatively, deep-learning-based methods train different architectures, such as two-stream CNN’s, 3D-CNN’s and recurrent networks, to detect violence in a video. Among these two approaches, deep-learning methods perform better by using large datasets during training.

### 2.1. Hand-Crafted Features

These methods use different descriptors such as histograms, interest points, trajectories, optical flow, and sparse codification. Methods based on histograms extract descriptors such as Histogram of Oriented Gradients (HOG), Histogram of Optical Flow (HOF), or a combination of these. Zhou et al. [7] extract local descriptors by segmenting motion regions using optical flow and Local HOG and Local HOF. Deb et al. [28] present a BoVW pipeline using Haar features, HOG, and HOF with Vector of Linearly Aggregated Descriptors (VLAD) [29] encoding to obtain robust video representations. Their method achieves high accuracy in the Hockey Fights and Movies datasets [30]. Still, the performance is poor in crowded scenes where many people are involved in violent action. An extension of HOG for three-dimensional data based on HOG3D is proposed by Yu et al. [31]. The HOG3D descriptors are extracted from video blocks; then, these descriptors are encoded following a BoVW framework. Finally, a violence classifier based on Kernel Extreme Learning Machine [32] is trained instead of an SVM. The main disadvantage of their method is the high computational cost due to the large dictionary (1000 visual words) used and the high cost of the HOG3D descriptor. Yu et al. [33] propose a novel violence-detection method based on HOG3D and MIL. They used a BoVW framework to build the final visual descriptor of a video by combining HOG3D features extracted from video clips. The MIL formulation is proposed to remove noise in training and testing data. Although their work outperforms other histogram-based methods, it still shows low performance compared to recent works.

### 2.2. Convolutional Neural Networks

Methods based on convolutional neural networks automatically learn a series of filters to extract hidden features. A typical approach followed in the literature is the two-stream approach, i.e., using two CNN’s. This architecture has been successfully used in various computer vision tasks, including violence detection [8,34], and spatiotemporal action-detection methods [3,19,21].

Xia et al. [8] present a two-stream 2D-CNN with an SVM classifier to detect violence in videos. First, they extracted video features using a *VGG-f* [35] model pretrained in *Imagenet* in each stream. The first stream extracts visual features from video frames, while the second extracts motion features from frame differences between consecutive frames. Then, two SVM classifiers are separately trained for each stream. The final detection is obtained using a label fusion method. Their method presents a low detection time; however, it does not detect violent actions between close persons in crowded scenarios. Meng and Yuan [34] propose a two-stream CNN architecture combined with improved trajectories to capture long-term information. Two VGG-19 networks extract spatial and temporal information from video frames and dense optical flow. Each network is fine-tuned separately, and improved trajectories are extracted from feature maps of the temporal stream. Their method presents better performance in crowded scenarios than in others.

Research works such as Cheng et al. [17] and Li et al. [12] replaced the conventional CNN with a 3D-CNN in each stream. Cheng et al. [17] present the Flow Gated Network model with four parts: the RGB stream, the Optical Flow stream, the Merging Block, and the final classifier. The RGB and Optical Flow streams are 3D CNN’s with consistent structures. The Merging stream is also a 3D-CNN used to process fused information from the previous streams. The Optical Flow stream helps build a self-learning pooling mechanism of the two streams by multiplying the output of the RGB stream by the output of the Optical Flow stream. Therefore, the Optical Flow stream is used as a gate to determine what information the model should preserve or drop. Their method achieves good performance in videos with a static camera. Therefore, low performance is presented in videos with variable illumination and camera movement. Li et al. [12] detect violence in crowded scenarios using two 3D-CNN’s trained over RGB frames and optical flow images. Although their method achieves high accuracy, it is computationally expensive during training.

Other methods use only one 3D-CNN to detect violent actions in video clips. Song et al. [11] modify a 3D-CNN by proposing a new preprocessing step based on keyframe selection. The frame selection method performs a uniform sampling for short videos and a keyframe selection for long videos. Li et al. [13] modify a 3D-CNN improving the internal architecture with dense layers to increase the capability of capturing spatiotemporal features of videos. These modifications improve the accuracy and computational efficiency by reducing the kernel size of convolutional filters and encouraging features reusing using dense layers [14].

Methods based on recurrent neural networks learn frame sequences to represent violent actions. Ditsanthia et al. [36] present a CNN with a bidirectional LSTM architecture. In their approach, first, they leverage features extracted from image-based CNN’s as ResNet-50 [37] to represent spatial information in a video frame. Then, a bidirectional LSTM is applied to learn a video-level violence classifier. Compared to other deep-learning-based methods, their method shows poor detection accuracy. Sudhakaran and Lanz [38] improve the architecture of Ditsanthia et al. [36], replacing the LSTM cells with convolutional LSTM cells and achieving higher accuracy. A combination of ResNet [37] and ConvLSTM is presented in Vosta and Yow [39] to detect violence events in real-world surveillance videos. They treat a violent action as an abnormal event, and the model is trained to classify the correct category for each anomaly. Traoré and Akhloufi [40] propose an end-to-end deep-learning architecture that combines a 2D-CNN with a bi-directional RCNN. The recurrent network refines inputs from two sources, optical flow and RGB frames, and a fully connected layer with sigmoid activation is used as a classifier. Islam et al. [41] leverage Separable Convolutional LSTM (SepConvLSTM) with a two-stream architecture to learn long-range spatiotemporal features. So far, this work has achieved state-of-the-art results in violence detection.

### 2.3. Spatiotemporal Violence Detection

As mentioned, most prior works [17,39,41] have focused on video classification tasks. However, progress in spatiotemporal violence detection has lagged. Table 1 summarizes our analysis of the literature on violence detection with respect to spatiotemporal detection. We can see that most of the methods only achieve temporal detection by classifying short clips as violent or non-violent. Only a few works such as Zhang et al. [42], Ribeiro et al. [43], Xu et al. [44], Choqueluque-Roman and Cámara-Chávez [45] address the problem of locating violent actions spatially and temporarily.

In Zhang et al. [42], the authors propose a spatiotemporal violence detector by generating candidate regions using optical flow and a Gaussian model called GMOF. Once obtained, the OHOF descriptor is extracted from 3D volumes densely sampled from the candidate regions. Spatial localization consists of classifying the 3D volumes using an SVM classifier. However, the authors do not report localization results, only classification accuracy. Ribeiro et al. [43] represent violent motion as unstructured motion using the RIMOC descriptor to learn statistic models of normal movements in a weakly supervised manner. Their method is robust to occlusions and complex backgrounds, but it presents a high false-positive rate in videos with variable illumination and shadows. The authors only report the classification accuracy. Choqueluque-Roman and Cámara-Chávez [45] propose a two-stage method to detect and localize violent actions in videos by using dynamic images [26] to extract violent motion information. In their work, the first step classifies a short video clip as violent or non-violent. Then, the localization step relies on salient regions extracted from the knowledge of a fine-tuned violence model, an object detector, and a refinement step. Although their method achieves similar results to the state of the art, the localization step is limited by the classifier performance and object detection. Xu et al. [44] present a two-stream localization guided framework to detect fights in surveillance videos. First, potential region proposals are extracted from motion activation maps. Then, human proposals are used to cluster the proposals through spatial relations between human proposals and motion regions. They use acceleration images extracted from optical flow magnitudes to train the temporal stream, and finally, a late fusion strategy is adopted to fuse predictions from both streams. Although spatial localization is performed in their pre-processing step, the human proposals are only used to guide the model to discard the background motion and achieve better classification performance.

### 2.4. Weakly Supervised Spatiotemporal Action Detection

Spatiotemporal violence detection is less studied due to the lack of video datasets annotated with spatiotemporal labels. The exorbitant cost of annotating each action frame-by-frame with bounding boxes has increased the need for research into weakly supervised methods, in order to reduce the annotation cost to only video-level labels. In this context, weakly supervised action-detection approaches and action-segmentation methods are explored to build our proposal to detect violent actions spatially and temporarily using only video-level labels.

Weinzaepfel et al. [56] propose a weaklysupervised action localization method by extracting human tubes using a human detector with a tracking algorithm. They propose a multi-fold MIL to select human tubes containing the action using the video-level labels. Each tube is represented using improved dense trajectories with Fisher vectors. Finally, an SVM is trained to classify tubelets by using multi-fold MIL. Weinzaepfel et al. [57] present an approach for spatiotemporal action localization using sparse spatial supervision, where the temporal extension of the action and one bounding box per instance are available during training. The first step of their approach is to extract human tubes relying on a human detector as Faster R-CNN [58] fine-tuned with additional data. Then, each tube is scored by a two-stream CNN combined with Improved Trajectories.

In Yan et al. [59], a multi-task ranking model for action–actor segmentation using only video-level labels during training is proposed. They generate action tubes and action–actor tubes by segmenting videos into supervoxels using a graph-based segmentation method. Action tubes are used as proposals for actions, e.g., walking, adult running, and crawling. CNN features are extracted from each tube to train the ranking model to select the most representative action tubes.

Arnab et al. [24] propose an MIL framework by leveraging person detectors pretrained on large image datasets. In their method, person tubelets are created by linking person detections over consecutive frames. A bag for MIL consists of all tubelets within a video, and it is annotated with the video-level label. During training, due to computational constraints, a whole bag cannot be processed simultaneously; therefore, they model the label noise through the uncertainty of sampling bags that do not contain any tubelets with the labeled action.

The method in Yang and Yuan [60] performs spatiotemporal localization of common actions in videos by using a two-step approach. The first step uses the Faster R-CNN [58] object detector to extract action proposals as action tubes, i.e., temporal sequences of bounding boxes, that locate action instances. The second step selects only common actions from the initial proposals by solving a subgraph density maximization problem.

Soomro and Shah [61] present an unsupervised action localization method by discovering action classes from unlabeled videos. They discover action labels using a discriminative clustering method with dominant sets. Then, they annotate training videos using a Knapsack approach given the discovered action classes. Finally, an SVM is trained as an action classifier to localize actions in testing videos. Mettes et al. [62] determine the spatiotemporal location of actions in a video by using pseudo-annotations. They exploit spatiotemporal pseudo-annotations from different sources such as action proposals, object proposals, person detections, motion, and center biases. These pseudo-annotations are combined using a correlation metric to train a classifier using MIL with only video-level labels. Chéron et al. [63] propose a spatiotemporal action-detection method with different levels of supervision. Their method is based on discriminative clustering to infer action labels for person detections by training a linear classifier with different constraints. Discriminative clustering assigns human tracklets to action labels by clustering data, and the clustering and classifier are simultaneously optimized.

Another approach for weakly supervised action detection is leveraging salient regions extracted from the video as action proposals or pseudo-annotations. In Chen et al. [64], the authors propose a two-stage weakly supervised video actor–action-segmentation framework with a new training methodology. In the first stage, frame attention maps are extracted from a pre-trained 3D-CNN to extract salient regions from a video sequence. These regions are then used as pseudo-annotations to initialize the segmentation network. In the second stage, the pseudo-annotations are refined using a selective train–predict cycle, where high-quality pseudo-annotations are selected in each iteration. Jian et al. [65] propose a video-saliency detection method that integrates object proposals with attention networks. In their method, the YOLOv3 detector [66] is used to extract salient region proposals as a pre-processing step. Then, three consecutive frames are fed to three Fully Convolutional Networks (FCN) with a weight-sharing mechanism to leverage the consistency of saliency maps between consecutive frames.

## 3. Proposed Method

The scope of our approach is to detect violent physical actions such as fights, arrests, and burglaries, and we use surveillance videos from fixed cameras. We leverage dynamic images [26] and image-based person detectors to generate spatiotemporal proposals or action tubes within an MIL framework to train a violence detector using only weak, video-level annotations. An action tube is defined as a set of bounding boxes linked over time, and each bounding box encloses the action, e.g., person walking or person fighting. These action tubes are used as instances in a video; all action tubes in the video form a bag. Figure 1 provides an overview of the proposed method. The method begins by generating action tubes in a video sequence. These tubes are obtained using a pre-trained person detector, motion segmentation, and an online linking algorithm. After that, each action tube is fed to a Fast-RCNN [25]-style convolutional neural network to extract spatiotemporal information. Then, all the action tubes are pooled/aggregated together. During training, the aggregated predictions of the tubes are compared with the video-level label. The next sections detail each stage of building action tubes followed by the model architecture.

### 3.1. Extracting Violent Action Tubes

In this part, our goal is to extract violent action tubes from a video. This process is divided into person detection, motion segmentation, and tube generation. The person detection extracts, frame-by-frame, spatial localizations (bounding boxes) of detected persons. The motion segmentation computes the spatial localizations of regions with movement (motion map) in the video. Finally, the tube generation combines the person detections and the motion map to temporally link the frame detections.

#### 3.1.1. Person Detection

To detect persons in a video frame-by-frame, we leverage a person detector trained on large image datasets, such as the Microsoft COCO dataset [67] and the Crowdhuman dataset [68]. We run a person detector as Chu et al. [69] on our training videos, and create person detections over consecutive frames.

#### 3.1.2. Motion Segmentation

In this part, we aim to obtain spatial motion regions in a video. First, we leverage dynamic images to discard background noise, and then we obtain a motion map with violent motion information using a non-motion suppression step. Dynamic images, compared to computationally expensive optical flow commonly used in the literature [17,34,42,43,44,52,70], have been less used in violence detection. A dynamic image is a standard RGB image that presents the appearance and dynamics of a video, with the main advantage of representing a sequence of frames without loss of information. In addition, dynamic images are an efficient approach to presenting video information to a CNN by summarizing the video content into a single still image which can be processed using standard CNN architecture [71]. To overload image pixels with long-term dynamic information, a dynamic image is constructed using the *Rank Pooling* method [72].

#### The Rank-Pooling Method

Given a video with $I_{1},I_{2},\cdots ,I_{T}$ frames, let $\varphi \left(I_{t}\right)\in R^{d}$ be a feature vector extracted from each individual frame $I_{t}$. First, a time average of these features up to time *t* is computed as: $V_{t}=\frac{1}{t}\sum_{T=1}^{t}\varphi \left(I_{t}\right)$. Then, the *ranking function* associates to each time *t* a score $S\left(t|d\right)=\langle d,V_{t}\rangle$, where $d\in R^{d}$ is a vector of parameters. These function parameters *d* are learned to reflect the rank of the frames in the video by the scores. Therefore, larger scores represent later times, e.g., ∀q,t if q>t then S(q|d)>S(t|d). The function parameters *d* are learned by using the *RankSVM* [73] formulation:$$d^{*}=\rho \left(I_{1},\cdots ,I_{T};\varphi \right)=\underset{d}{\mathrm{argmin}}E\left(d\right)$$
(1)$$E\left(d\right)=\frac{\lambda }{2}{∥d∥}^{2}+\frac{2}{T(T-1)}\times \sum_{q>t}\max\left\{0,1-S\left(q|d\right)+S\left(t|d\right)\right\}$$

The first term in Equation (Equation 1) is the usual quadratic regularizer used in SVMs. The second term is a hinge-loss soft-counter to register how many pairs q>t are incorrectly ranked by the scoring function. A pair is correctly ranked only if at least a unit margin separates the scores, e.g, S(q|d)>S(t|d)+1. The function $\rho (I_{1},\cdots ,I_{T};\varphi )$ is defined by optimizing Equation (Equation 1) to map a sequence of *T* video frames to a single vector $d^{*}$. This vector contains information to rank all the video frames by aggregating information from all of them. The process of constructing $d^{*}$ from a sequence of video frames is called *Rank Pooling*.

In contrast to the work presented in Fernando et al. [72], Bilen et al. [71] apply rank pooling directly to the RGB image pixels instead of using the local features (HOG, HOF, etc.) as the feature vector φ(.) (extracted from each frame). Therefore, the $\varphi (I_{t})$ is now an operator that stacks RGB components of each pixel in image $I_{t}$ on a large vector. In this case, the vector $d^{*}$ is a real vector that has the same number of elements as a single video frame, and it can be interpreted as a standard RGB image called *dynamic image*.

#### Approximate Rank Pooling

As we have described before, constructing a dynamic image entails solving Equation (Equation 1). Solving this optimization problem could be simplified by an approximation technique called *approximate rank pooling* [71]. The approximate rank-pooling method is a weighted combination of the data points $\varphi (I_{t})$ indicated by Equation (Equation 2).
(2)$$\hat{\rho }\left(I_{1},\cdots ,I_{T};\varphi \right)=\sum_{t=1}^{T}\alpha_{t}\varphi \left(I_{t}\right)$$
where the coefficients $\alpha_{t}$ are given by:(3)$$\alpha_{t}=2\left(T-t+1\right)-\left(T+1\right)\left(H_{T}-H_{t-1}\right)$$
where $H_{t}=\sum_{i=1}^{t}1/t$ is the t−th Harmonic number and $H_{0}=0$. Hence, the approximate rank-pooling method accumulates the video frames after pre-multiplying them by $\alpha_{t}$.

Figure 2 presents some examples of dynamic images summarizing violent actions. Generally, violent actions present fast movement between individuals; therefore, dynamic images present key steps of the action (shadows), and they capture motion through time with motion blur (lower right image in Figure 2).

#### Violent Spots

As we can see in Figure 2, a dynamic image highlights pixels where there is high motion. Based on visual analysis, we observed that the brightest and darkest spots are near spatial locations where there is fast movement. We consider these locations as regions with a violent motion that we will call *violent spots*.

To extract these spots, given an input video, we break it into short video clips of duration τ (τ=5 in our experiments) to create multiple short video clips per video. Thus, an input video with *T* frames produces s=T/τ short clips. Then, for each video clip, we construct a dynamic image following Equation (Equation 2). Next, we look for the *violent spots* using thresholds. Figure 3 shows this process graphically.

#### Non-Motion Suppression

To suppress non-violent motion frame-by-frame, we first binarize the dynamic image by normalization. This binarized image suppresses background motion. We then multiply this binarized image by every frame in the clip, which accentuates the motion regions in the frame by suppressing non-moving regions. Finally, we apply a connected component analysis algorithm to detect connected regions. Among these regions, we choose regions that have most of the brightest and darkest spots (violent spots) found previously. Figure 4 shows this process graphically. These regions are finally used in the next steps to construct action tubes.

#### 3.1.3. Tube Generation

The tube generator is an online linking algorithm adapted from Singh et al. [74] to incrementally construct action tubes using the person detections and the motion map extracted in previous steps. Given a set of detections at time t=1…T, we seek the set of consecutive detections (action tube) $T=\{b_{ts},\ldots ,b_{te}\}$, where $b_{ts}$ is the first detection of the tube and $b_{te}$ the detection at the end. These tubes are more likely to constitute an action instance among all possible such collections. An action tube is allowed to start or end at any given time because the algorithm incrementally (frame-by-frame) builds multiple action tubes in parallel. Similarly to Singh et al. [74], an online algorithm is used to associate detection boxes in the upcoming frame with the current set of (partial) action tubes. Figure 5 shows a 3D volumetric view of a video showing detection boxes and selected frames.

Algorithm 1 presents the steps to construct the action tubes. We build action tubes following the next steps for a given input video *V* with its frames $I_{1},I_{2},\ldots ,I_{T}$.

Tube Initialization. The first step to initialize an action tube is to compute the motion map using the previous segmentation step. This motion map presents motion regions in the τ first frames. In the meantime, person detections are extracted by running the pre-trained object detector. Then, we look for close persons who could be interacting between them. We merge person detections (merged detections) if the IoU of them is greater than a threshold $\lambda_{1}$. Then, an action tube is initialized in spatial locations where there are persons and motion, i.e., we initialize an action tube in the merged detections that have IoU greater than a threshold $\lambda_{2}$ with a motion region (merged-detections motion).Temporal Linking. We update the current set of partial action tubes with the current merged-detection motion to build an action tube incrementally. To update an action tube $T_{i}$ with a merged-detection motion, we calculate the IoU between the last box of $T_{i}$ and a merged-detection motion $b_{j}$. The action tube $T_{i}$ is updated with $b_{j}$, i.e., $b_{j}$ is assigned to $T_{i}$ if the IoU between them is greater than 0.4. We retain the non-updated tubes unless more than $t_{k}$ frames have passed with no match found for them. An action tube is finished if $t_{k}$ frames have passed without an update. If any merged-detection motion is left unassigned, we start a new action tube using this region.

**Algorithm 1** Tube Generation**Input:** video V={I1,I2,…,IT}**Output:** action tubes lp={T1,…,Ti}
lp←∅                                                                                              ▹ Initialize list of action tubes.M←{b1,…,bl}                                            ▹ Compute motion map by using all frames It.**for**t←1 to *T*
**do**                                                                               ▹ Traverse the video frames.    B←{bj,j=1…n}                                            ▹ Compute person detections at frame *t*.    I←{IoU(bj,bh),j,h=1…n}   ▹ Compute the IoU between all the person detections.    Bm←∅                                                                                                      ▹ Merged detections.    **for**
*i* in *I* **do**        **if** i≥λ1**then**                                                                                                  ▹ Close persons.           Bm←bj∪bh                                                                  ▹ Add merged close detections.        **else**           Bm←bj           Bm←bh        **end if**    **end for**    I←{IoU(bj,M),j=1…n}        ▹ Compute the IoU between Bm and motion map.    **for**
*i* in *I* **do**        **if** i≥λ2
**then**                                                              ▹ Regions with persons and motion.           **if** lp is empty **then**                                                                              ▹ Start new tube at *t*.               Bm←bj           **else**                                                        ▹ Find a matching tube for the current detection.               **for** Ti in lp
**do**                   iou←IoU(bj,Ti)                                             ▹ IoU between detection and tube.                   **if** iou≥0.4
**then**                       Ti←bj                                               ▹ add/update tube with current detection.                   **end if**               **end for**           **end if**        **end if**    **end for****end for**


### 3.2. Violence-Detection Model

In order to detect violence, we propose an end-to-end convolutional neural network inspired by I3D [75] and Faster R-CNN [58]. Similarly to recent approaches for spatiotemporal action localization such as Gu et al. [19], Girdhar et al. [20], Arnab et al. [24] and Köpüklü et al. [21], we propose a two-stream CNN illustrated in Figure 6, which can be divided into five major parts: 3D-CNN branch, 2D-CNN branch, fusion module, tube pooling, and classifier.

#### 3.2.1. 3D-CNN Branch

For video understanding tasks, the most critical issue is how to extract representative features of the spatiotemporal information. In this context, 3D CNNs can simultaneously capture spatiotemporal features by applying convolution operations in space and time dimensions. Similarly to Gu et al. [19] and Girdhar et al. [20], we utilize the Inception 3D (I3D) [75] architecture to extract spatiotemporal information from an input video. In this branch, we modified the I3D architecture by discarding the layers after the *Mixed 4e* layer. The input is a *tube clip* of the video, i.e., a sequence of successive central frames in time order extracted from a tube constructed in previous steps. For instance, in Figure 5, the action tube 1 (red) has a length of 40 frames, therefore, we select D=16 central frames (from frame 12 to frame 28) as the *tube clip*. This clip has a shape of [C×D×H×W], where C=3, *D* is the number of input frames, and *H* and *W* are the height and width of the input images. The output of the modified I3D is a feature map of shape $\left[C^{\prime }\times D^{\prime }\times H^{\prime }\times W^{\prime }\right]$, where $C^{\prime }$ is the number of output channels, $D^{\prime }=4$, $H^{\prime }=\frac{H}{16}$ and $W^{\prime }=\frac{W}{16}$. Finally, this feature map is passed through a 2D RoI Pooling layer extended to 3D (3D RoI Pool) by applying the 2D RoI Pooling at the same spatial location over all time steps, similar to Gu et al. [19] and Feichtenhofer et al. [22]. The RoI pooling layer takes the output feature map to project the central bounding box of the tube (Tube central box) to obtain region features. The output of the RoI pooling layer is the feature map of shape $\left[C^{\prime }\times D^{\prime }\times H^{\prime \prime }\times W^{\prime \prime }\right]$ that is temporally pooled to obtain the final shape of $\left[C^{\prime }\times 1\times H^{\prime \prime }\times W^{\prime \prime }\right]$.

#### 3.2.2. 2D-CNN Branch

Meanwhile, to obtain spatial information, we extract 2D features from one frame (tube keyframe). We employ ResNet-50 [37] as the basic architecture in this branch. The input keyframe with the shape [C×H×W], where C=3, *H* and *W* are the height and width of the keyframe, respectively, is passed through the ResNet-50 up to *Layer 3*, resulting in a feature map of shape $\left[C^{\prime \prime }\times H^{\prime }\times W^{\prime }\right]$, where $C^{\prime \prime }$ is the number of output channels, $H^{\prime }=\frac{H}{16}$ and $W^{\prime }=\frac{W}{16}$. Like the 3D branch, the output feature map is passed through a 2D pooling layer to obtain region features with shape $\left[C^{\prime \prime }\times H^{\prime \prime }\times W^{\prime \prime }\right]$.

#### 3.2.3. Channel Fusion and Attention Mechanism (CFAM)

To fuse both branches, we use the CFAM module [21] which is based on Gram matrix [76] to map inter-channel dependencies, and it is beneficial for fusing features coming from different sources [21]. CFAM fuses the feature maps using concatenation, stacking the features along channels to model interdependencies between channels. By exploiting the interdependencies between channel maps, we could emphasize interdependent feature maps and improve the feature representation of specific semantics [77]. First, the output of both 3D and 2D branches are of the same shape in the last two dimensions, such that these feature maps can be easily concatenated along channels. As a result, the fused feature map *A* has a shape of $\left[C^{\prime }+C^{\prime \prime }\times H^{\prime \prime }\times W^{\prime \prime }\right]$, passing it to the CFAM module.

The CFAM module feeds *A* into two convolutional layers to generate the feature map *B* of shape $\left[C\times H^{\prime \prime }\times W^{\prime \prime }\right]$. Then, *B* is reshaped to the tensor $F\in R^{C\times N}$, where $N=H^{\prime \prime }\times W^{\prime \prime }$. Then, the Gram matrix *G* is computed to obtain the feature correlations across channels, following the next equation:(4)$$G=F\cdot F^{T}\;\mathrm{with}\;G_{ij}=\sum_{k=1}^{N}F_{ik}\cdot F_{jk}$$
where $F^{T}$ is the transpose of *F*. After computing *G*, a softmax layer is applied to generate channel attention map $M\in R^{C\times C}$:(5)$$M_{ij}=\frac{\exp\left(G_{ij}\right)}{\sum_{j=1}^{C}\exp\left(G_{ij}\right)}$$
where *M* summaries the inter-channel dependency of features given a feature map. Next, a matrix multiplication between *M* and *F* is performed, and the result is reshaped to the same shape as the input tensor $\left[C\times H^{\prime \prime }\times W^{\prime \prime }\right]$, resulting in the tensor $F^{\prime \prime }$. Finally, the output of the CFAM module is computed by combining $F^{\prime \prime }$ and the original input feature map *B* with a trainable scalar parameter α using an element-wise sum operation, and α gradually learns a weight from 0:(6)$$C=\alpha \cdot F^{\prime \prime }+B$$
where *C* is the final feature of each channel, and it is a weighted sum of the features of all channels and original features. Finally, the feature map *C* is fed into two more convolutional layers to generate the output feature map *D* with shape $\left[C^{*}\times H^{\prime \prime }\times W^{\prime \prime }\right]$. Convolutional layers at the beginning and end of the CFAM module help mix features coming from different backbones with possibly different distributions.

#### 3.2.4. Tube Pooling

So far, for a given input video with *k* tubes (k=3 in Figure 6), we passed each tube through the two-stream model to obtain *k* feature maps (one for each tube) because each tube contains different spatial and temporal information in the video. To fuse these *k* feature maps into one, we perform an element-wise max pooling operation.

#### 3.2.5. Classifier

The fused feature map from the pooling layer is passed through a final convolutional layer with 1×1 kernels to generate the desired number of output channels. For binary classification, the number of output channels is 2, and the model is trained to classify an input video as violent or non-violent.

### 3.3. Spatiotemporal Violence Detection

In order to train an instance-level classifier, i.e., a tube classifier, we rely on an MIL framework similar to in Arnab et al. [24]. A bag for MIL consists of all tubes within a video and is annotated with the video-level labels (violence and non-violence) that we have as supervision. The size of the bag is determined by the number of tubes generated by the tube generator (detailed in Section 3.1). Therefore, the spatiotemporal violence detection is factorized, i.e., the spatial localization capability of the model depends on the quality of the person detections, and the tube generator performs the temporal localization. Figure 7 shows the tube classifier architecture. We slightly modify the proposed architecture detailed in Section 3.2 by removing the tube pooling layer and changing the kernel outputs of the classifier to 1. The outputs of the classifier are passed through a Sigmoid activation function to obtain tube-level probabilities. To train the tube classifier, we use an aggregation function as *max-pooling* to aggregate the tube-level probabilities (instance probabilities) to obtain the video-level probability (bag probability) [24]. Once we have the video-level probabilities and the video-level ground truth, we train the model by optimizing the binary cross-entropy loss.

## 4. Experiments and Results

This section presents the experiments that validate our proposal to detect violence. We evaluate our method on three public benchmark datasets: *Hockey Fight* [30], *RWF-200* [17], *RLVS* [78], and *UCFCrime2Local* [79]. The first three datasets are designed particularly to evaluate the classification performance of violence, and the last one is used to evaluate the localization performance. Although there are other datasets for violence detection [80,81], they have high heterogeneity in data format, annotation type, data source, etc., which makes it difficult to evaluate a method with them. Therefore, we only evaluate our proposed method on these four datasets because they are all consistent with video surveillance data, with similar format, annotation type, data source, and close video footage.

Hockey Fight [30] was designed to evaluate violence detection methods with only a few persons in the action. The videos were extracted from hockey games from the *National Hockey League (NHL)*. The dataset contains 1000 videos of short duration (1 or 2 s), where 500 of these are fights and 500 non-fight. Each video consists of 40–50 frames of 720×576. All videos are relatively uniform in duration and content, and only video-level labels are provided.

RWF-2000 [17] consists of 2000 video clips captured by surveillance cameras in real-world scenes. Half of the videos include violent behavior, while others belong to non-violent activities. All videos were captured from the perspective of surveillance cameras, i.e., they are similar to real violent events. The dataset is split into two parts: the training set (80%) and the test set (20%), and only video-level labels are provided.

RLVS [78] is a benchmark of 2000 videos divided into 1000 violence clips and 1000 non-violence clips. The violent clips include fights in different environments such as streets, prisons, and schools. The non-violent clips contain other human actions such as playing basketball, tennis, and eating. The videos have a variable duration of between three and seven seconds. All videos have a high resolution of 397×511 on average.

UCFCrime2Local [79] is a dataset designed particularly to evaluate anomaly detection methods and is a subset of the UCF-Crime dataset [3] with spatiotemporal annotations in its training set. The dataset presents six human-based anomaly categories such as *Arrest*, *Assault*, *Burglary*, *Robbery*, *Stealing*, and *Vandalism*. The dataset contains 100 abnormal videos and 200 normal videos. All videos have a long duration (40 s) with 2 or 3 abnormal instances per video. We use this dataset to evaluate the spatiotemporal detection of violent actions.

### 4.1. Evaluation Metrics

In this section, we briefly explain the evaluation metrics we use, according to the literature, to evaluate the performance of the proposed method.

Classification Accuracy: Similarly to other tasks, the classification accuracy in violence detection is one way to measure how often a method classifies a video correctly. More formally, it is defined as the number of true positives (TP) and true negatives (TN) divided by the number of true positives (TP), true negatives (TN), false positives (FP), and false negatives (FN), see Equation (Equation 7). This metric works well if there are an equal number of samples belonging to each class, and most of the works in the literature use this metric to evaluate the performance.
(7)$$Acc=\frac{TP+TN}{TP+TN+FP+FN}$$Area Under ROC curve (AUC): The AUC determines how much a classifier is capable of distinguishing one class from another one. It is the area under the Receiver Operating Characteristics (ROC) curve, which is a graphical plot created by plotting True Positive Rate (TPR) against False Positive Rate (FPR).
(8)$$TPR=\frac{TP}{TP+FN}$$
(9)$$FPR=\frac{FP}{FP+TN}$$Localization Error: The localization error measures the error rate of spatial localization at the frame-level, commonly used in weakly supervised object detectors [82,83]. Given a predicted box, it has to have an *Intersection over Union* (IoU) greater than 0.5 with the ground-truth bounding box to consider the localization successful, see Equation (Equation 10). Otherwise, it is counted as an error.
(10)$$IoU=\left|\frac{A\cap B}{A\cup B}\right|$$
where *A* and *B* are two arbitrary convex shapes. For simplicity, it is adopted box shapes.

### 4.2. Experimental Settings

#### 4.2.1. Tube Generation

To generate action tubes, we use person detections obtained from Chu et al. [69] pretrained with Crowdhuman dataset [68]. This model allows us to detect multiple persons heavily overlapped, which is a relevant characteristic in violent scenes where two or more people interact. We merge highly overlapped person detections using Algorithm 1 with $\lambda_{1}=0.3$, and we only track these merged detections. To track only merged detections with movement, we look for merged detections with IoU between the merged detection and the motion region greater than $\lambda_{2}=0.4$ (merged-detections motion). Note that we only track this merged-detections motion, i.e., we start a new action tube in each merged-detection motion because it is more likely to constitute a violent instance. We use a temporal window of τ=5 frames of duration per clip during motion segmentation. Then, we construct a dynamic image per clip using *Approximate Rank Pooling*. Each dynamic image is equalized using the *k-means* algorithm with k=5 to reduce the number of colors in the image. Next, each equalized dynamic image is normalized using per-channel normalization using the mean and the standard deviation of the image. For connected component analysis, we use the *OpenCV* implementation of the SAUF (Scan Array-based Union-Find) algorithm [84].

#### 4.2.2. Model Training

We trained the proposed model with a batch size of 4, with four action tubes per video and 16 frames per tube. These frames were extracted from each tube using uniform sampling and resized to 224×224, resulting in a clip of shape 16×224×224×3, and this clip is fed to the 3D branch. For the 2D branch, we compute the dynamic image corresponding to the clip resulting in an image of shape 224×224×3, i.e., we use a dynamic image as a keyframe.

The proposed model was implemented using the Pytorch library [85], and the experiments were performed using Google Colab notebooks with 12 GB of memory and a GPU Tesla T4. We trained the model for about 100 epochs or until the model started overfitting. The 3D branch was initialized using weights pre-trained on the Kinetics [86] dataset, and the 2D branch was initialized using weights pre-trained on the ImageNet [87] dataset. For the CFAM module, we used the Xavier method to initialize the weights. For model optimization, we used Stochastic Gradient Descend (SGD) with a learning rate of $10^{-5}$, and we used the binary cross-entropy loss.

### 4.3. State-of-the-Art Comparison

We follow the official evaluation metrics strictly to report and compare the results with the state of the art. We report the classification accuracy for the *Hockey Fight*, *RLVS*, and RWF-2000 datasets and the localization error for *UCFCrime2Local* dataset.

#### 4.3.1. Experiments on Hockey Fight Dataset

Table 2 shows the comparison of the proposed method with previous works of the state of the art on the *Hockey Fight* dataset. We adopted five-fold cross-validation for this dataset to evaluate the classification performance. The top half of the table lists methods based on hand-crafted features, and deep learning-based methods are listed in the bottom half. We can see that deep-learning methods outperform earlier methods based on hand-crafted features. Among all of them, our method achieves results that are competitive with the best state-of-the-art results.

#### 4.3.2. Experiments on RWF-2000 Dataset

Table 3 presents the comparisons between the proposed method and other previous works. All methods report the classification accuracy on 20% of the dataset, and all use the remaining 80% for training. We obtain an accuracy of 88.45% on this dataset, achieving very close results to those of the the state of the art achieved by Islam et al. [41]. The proposed method achieves comparable performance in detecting violent actions in surveillance videos. Action tubes give the model spatiotemporal information as a pre-processing step.

#### 4.3.3. Experiments on RLVS Dataset

Table 4 presents the comparisons of the proposed method and other previous works that report the validation accuracy. Our proposed method achieves an accuracy of 92.88%, almost three points below the state of the art achieved by Soliman et al. [78]. We conjecture that this poor performance is because the dataset contains video footage with crowded scenes where many people are interacting. This causes poor performance of the action tube generator, and thus violent actions are more difficult to be spatially detected.

#### 4.3.4. Experiments on UCFCrime2Local Dataset

We report the localization error metric [45,82]. In our experiments, we only used categories with violent actions such as *Arrest, Assault, Robbery*, and *Stealing*. We leveraged the temporal annotations to extract short clips with violent and non-violent actions. Then, we trained the proposed instance-level classifier (Section 3.3) using 80% of the videos, and the evaluation was performed in the 20% remainder part. Therefore, the dataset was reduced to a few clips that could cause overfitting. To overcome this problem, we first trained the model on the RWF-2000 dataset, and then we used the weights of the trained model to initialize the training on this dataset. Table 5 presents the localization performance.

### 4.4. Ablation Studies

We conduct ablation studies of the proposed two-stream model on the RWF-2000 dataset. We experimented with three fusion strategies to combine the two streams. These strategies produce some variations of the proposed model that are detailed below. We also ablate the motion representation of violent actions using Optical Flow-based images and Dynamic Images.

#### 4.4.1. Fusion Strategies

We detail below the strategies that we use to fuse information from the two streams.

Late Fusion: this strategy is the most used because of its simplicity. Each branch is trained separately, and, at test time, the predictions of both branches are fused by calculating the average.Concatenation: the output feature maps of both streams are concatenated along the channel axis and passed through a classifier.CFAM: in this strategy, we use the CFAM module to fuse feature maps coming from different sources.

#### 4.4.2. Optical Flow or Dynamic Images

Table 6 ablates different variations of the proposed model. The most naive two-stream model is the proposed in Carreira and Zisserman [75] for action recognition, which uses two 3D-CNN’s, an RGB branch, and an Optical Flow (OF) branch with the Late Fusion strategy. As shown in the first row, this model performs the worst. Another variation is presented in the second row, where we replace the optical flow branch with a dynamic image (DI) branch with the Late Fusion strategy. We can see that dynamic images improve the model performance because they are more robust in representing long-range motion. In addition, dynamic images highlight the violent motion in the frames by suppressing non-moving backgrounds.

#### 4.4.3. Using Two 3D-CNN’s or a 3D-CNN with a 2D-CNN

So far, we have used a 3D-CNN for each branch. In the next models, we replace the second branch with a 2D-CNN to reduce the computational complexity of the model, as is proposed in Köpüklü et al. [21]. We feed 16 frames per tube to the 3D-CNN, and the dynamic image is constructed using these frames to the 2D-CNN. This experiment evaluates two fusion strategies: concatenation and the CFAM module. Table 6 (third row) shows that using a 2D-CNN with the concatenation strategy achieves a similar performance to previous models. However, concatenating the feature maps from both streams cannot provide a satisfying result. To overcome this problem, we use the CFAM module to fuse both streams, which improves the classification performance (fourth row) by around 2% compared to previous models. The explanation for this is that the attention mechanism in CFAM helps aggregate features from different sources better than concatenation and late fusion strategies.

#### 4.4.4. The Importance of Action Tubes

The fourth and fifth rows in Table 6 show the importance of the action tubes in the proposed model. Using the proposed two-stream model with CFAM produces poor results when no action tubes are used (fifth row). The best performance is achieved using the proposed two-stream model with action tubes (fourth row). Action tubes improve performance because temporal information in a video is filtered by discarding irrelevant frames for violent action analysis, e.g., frames without persons. In addition, the tube’s spatial regions help the model to focus on the action. With these results, we validate the contribution of the action tubes extracted by our method to improve the classification performance to detect violence.

#### 4.4.5. Number of Action Tubes per Video

During training, we ablate the number of action tubes sampled per video using the best variation of the proposed model (fourth row in Table 6). As shown in Table 7, our tube generator produces, on average, two or three action tubes per video. However, some videos have more than five tubes; in such cases, due to memory limitations, we have to sample up to four tubes per video because this is the maximum of tubes we can fit onto a 12 GB Tesla T4 GPU. We experimented with two sampling strategies: random sampling and motion-based sampling. In the first case, we randomly select four action tubes. In the second one, we train a 2D-CNN classifier using the Hockey Fight dataset to learn violent motion information from video footage. Next, we use the trained model to obtain a score for each action tube. Finally, we choose the top four action tubes to train the violence model.

Figure 8 presents some examples of videos where more than one tube was generated. We can see that the proposed method generates up to five action tubes in the second video (second row in Figure 8) because many persons were detected in the scene. However, at least one tube in each video contains a violent action. In general, during our qualitative analysis, we observed that an action tube is started in locations where two or more people are very close and there is fast movement in the same locations. The first row in Figure 8 presents a video with three action tubes that were extracted because persons move fast during the action.

Table 8 ablates the number of action tubes used during training. As expected, we observe that the model trained with four action tubes performs better because the positive instance is more likely to be sampled in videos where more than one tube was extracted. In addition, using motion scores to sample the action tubes slightly improves the classification performance.

#### 4.4.6. Number of Frames per Tube

As we have explained in Section 3.2, in the proposed method, for the 3D-CNN branch, the input is a sequence of successive frames from the tube (*tube clip*). In 3D-CNNs, different clip lengths can change the overall performance of the two-stream model [21,91]. Therefore, we conduct experiments with a different number of frames per tube. Table 9 shows the classification accuracy using different clip lengths per tube. We observe that the model with an input of 16 frames per tube performs better than an input of 8 frames since it contains more temporal information. However, as the number of frames increases, the classification performance worsens. We conjecture that a long sequence of frames may contain unrelated frames, which could add noise to the violent action. In addition, adding frames to a dynamic image leads to capturing background noise.

### 4.5. Qualitative Analysis

#### 4.5.1. Action Tubes

Figure 9 illustrates action tubes extracted using the proposed method. Each row presents a different violent video from the *UCFCrime2Local* dataset [79]. Each image shows a video frame with person detections (white boxes), ground truth (light blue boxes), and the action tubes extracted by our method (red boxes). Our approach localizes violent tubes well when persons are correctly detected. In some videos, such as the first one (first row), our method extracts up to three action tubes; however, at least one contains the violent action.

Nevertheless, our approach presents some failures. As mentioned, our method depends on the quality of the person detections to extract action tubes. Therefore, we first investigate the effectiveness of the tube generator, qualitatively analyzing the performance in videos where no person is detected, or only one person is detected. In these two cases, the method fails to generate action tubes. The first row in Figure 10 illustrates two localization errors: (1) the person detector only localizes one person because persons are strongly overlapped (Frame:1) and (2) missed detections of persons in unusual poses (from Frame:5 to Frame:20). These problems are due to the significant domain gap between the dataset used to train the person detector, Crowdhuman [68], and video surveillance datasets, e.g., RWF-2000 [17].

To handle these problems, we leverage the motion regions extracted from dynamic images to build action tubes during training. The second row in Figure 10 illustrates the motion regions (green boxes) in the video. The pink boxes are the motion regions with fast movement, i.e., regions with violent spots (Section 3.1.2). We track these motion regions to build an action tube when no person is detected in the video.

#### 4.5.2. Violence Localization

During testing, we feed all action tubes for a given video to obtain a violence score per tube. Then, we label the tubes with a score higher than a threshold of 0.5 as violent tubes. Figure 11 presents qualitative results of the proposed model for localizing violent actions. This figure presents successful cases where our method obtains high scores for well-extracted action tubes. Our method can localize violence in different situations with different camera positions, video footage, and numbers of people participating in the action. The first and fourth rows in Figure 11 show the localization of the violent action even with other people and motion in the scene.

Figure 12 shows some cases where the proposed method fails to localize violent actions. The first row shows a failure case since the first frames representing the violent actions are not localized. The persons were not detected due to the occlusion and camera position. The second row shows a different failure case where only one instance of violence is detected, and other people are not doing the same action in the scene. The third and fourth rows show false-positive detections where the model labels normal actions as violence. Most of these failure cases are because of poor performance in the tube-generation step. Note that for our tube generation, the action tubes are constructed using the detections obtained from the person detector that has been trained on a dataset with a significant domain gap with surveillance footage. These discrepancies produce missed detections where people with different poses and those who are highly occluded are not detected.

### 4.6. Time Analysis

Table 10 shows the time for each stage of the proposed method. The person detector [69] can extract bounding boxes within 0.0331 s per frame. The proposed tube-generation algorithm can process 16 frames in 0.0061 s, and the two-stream model can process a short video of 64 frames (4 tubes of 16 frames) within 0.199 s.

Since video surveillance analysis should work with real-time streaming frames, our method processes short clips of 64 frames. The person detector [69] and the tube generator does not perform at every frame. More specifically, the person detector network detects people every two frames, and the coordinates of the detected people are used as the input in the tube generator. The reason for not using detection in every frame is that it takes a lot of time in crowded scenarios. Since person detection is not performed at every frame, the tube generator is updated according to the detected people, and it can update multiple tubes quickly. Finally, the proposed two-stream model can infer four action tubes of 16 frames (224×224) within 0.1992 s. The inference time is obtained by following the benchmark script proposed by Zhao et al. [92], where we only care about the model inference time, excluding the IO time and pre-processing time. We used Google Colab and set the batch size (videos per GPU) to 1 to calculate the inference time.

## 5. Conclusions

In this work, we have presented a method to detect, spatially and temporally, violent actions in surveillance video. We presented a Fast-RCNN-style model to leverage spatial and temporal information from the video. We have proposed to extract action tubes by leveraging pre-trained person detectors and dynamic images as a pre-processing step to train a two-stream model to learn spatiotemporal features from violent actions. In order to label each action tube, we followed a Multiple Instance Learning (MIL) framework to train the model only with video-level labels. Our proposed method achieves results that are competitive with those of the state of the art on four public benchmarks, namely Hockey Fight, RLVD, RWF-2000, and UCFCrime2Local. Unlike previous works aiming to classify video clips, our method can also localize violent actions spatially using only video-level labels during training. In addition, we have conducted ablation studies to validate the fusion strategy for the two-stream model, the motion representation, and the effect of the number of action tubes sampled during training. Our results show a better performance using a two-stream model with 3D-CNN and 2D-CNN branches. The 3D branch learns spatiotemporal features from short clips, and the 2D branch learns important spatial features from a motion image (dynamic image). The best fusion strategy was an attention-based module that learns inter-channel dependencies with significant semantics. However, our approach presents some failures when the action tubes are not extracted correctly due to missed person detections and person occlusions. These errors cause the model to incorrectly classify and localize an action. In future works, we believe that improving the tube-extraction method can help us to achieve better results in spatiotemporal violence detection.

## Figures and Tables

**Figure 1 sensors-22-04502-f001:**
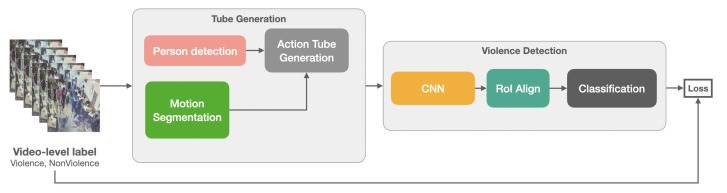
Overview of the proposed method.

**Figure 2 sensors-22-04502-f002:**
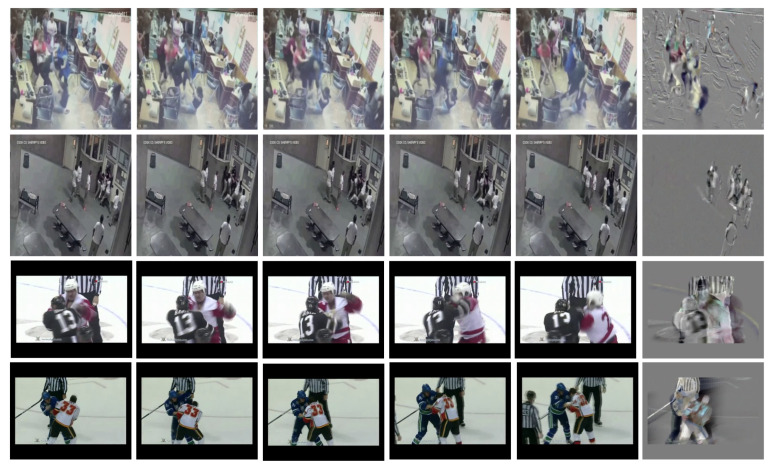
Examples of dynamic images representing violent actions. Each row represents a frame sequence of a video summarized by a dynamic image (last column).

**Figure 3 sensors-22-04502-f003:**
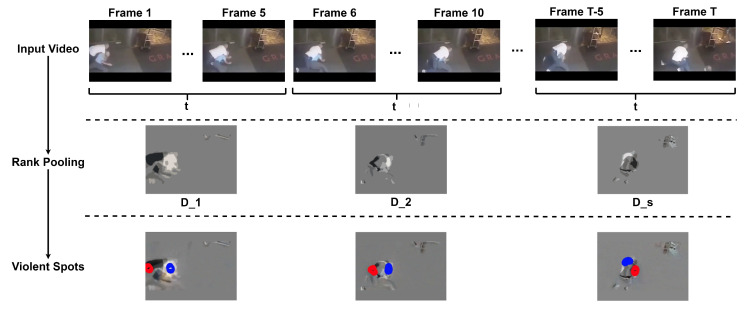
Violent Spots. First, an input video is broken into *s* clips of duration τ (first row). Each segment is summarized into a dynamic image to highlight motion regions in the segment (second row). The third row shows violent spots in the dynamic image to obtain regions with fast movement. The brightest spots are in blue, while the darkest spots are in red.

**Figure 4 sensors-22-04502-f004:**
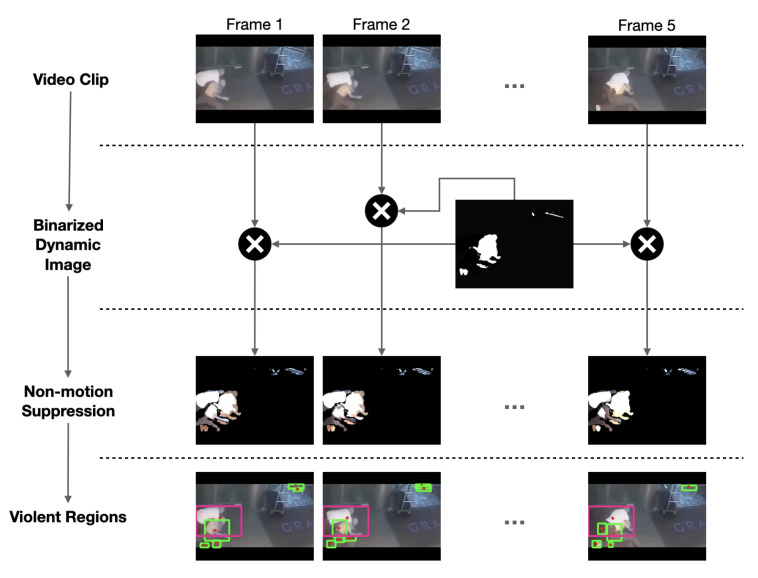
Non-motion Suppression. Each frame in the clip is multiplied by the binarized dynamic image. The third row shows the frames with non-moving regions suppressed. The fourth row shows the connected components (green boxes). The components with more movement (violent) are the pink boxes.

**Figure 5 sensors-22-04502-f005:**
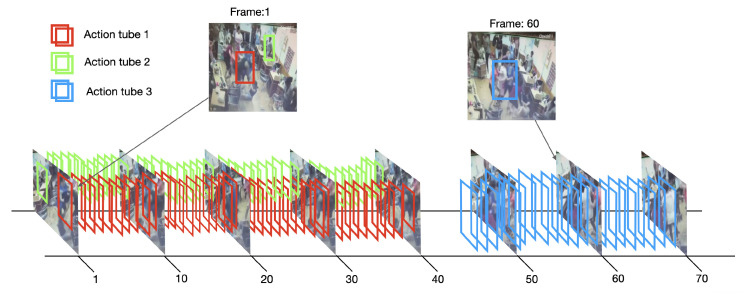
Online tube generation in a violent video from RWF-2000 dataset [17]. The detection boxes are incrementally linked to built space–time action tubes. Note that each tube could have a different temporal length, e.g., Action tube 1 has a length of 40 frames (from frame 1 to frame 40), and Action tube 3 has a length of 25 frames (from frame 45 to frame 70).

**Figure 6 sensors-22-04502-f006:**
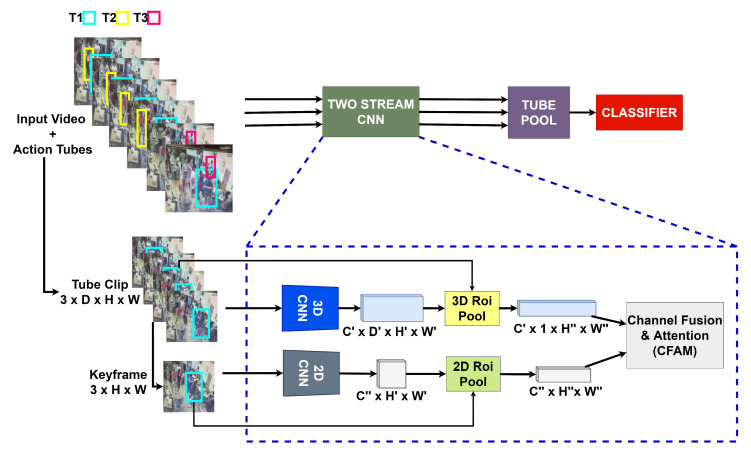
The proposed model is composed of two (3D-CNN + 2D-CNN) streams. After building action tubes (T1,T2 and T3), each *tube clip* is passed through the two-stream model. The dashed box shows the internal structure of the model. In the spatiotemporal branch, the *tube clip* is passed through the I3D, and the central tube box is projected to the output feature map. In the spatial branch, the keyframe of the *tube clip* is passed through a 2D-CNN, and the frame box is projected to the output feature map. Both branches are fused in the CFAM module. The fused feature maps of the two-stream model are pooled (using max pooling) to merge the feature maps into one. Finally, the pooled feature map is passed to the classifier network.

**Figure 7 sensors-22-04502-f007:**
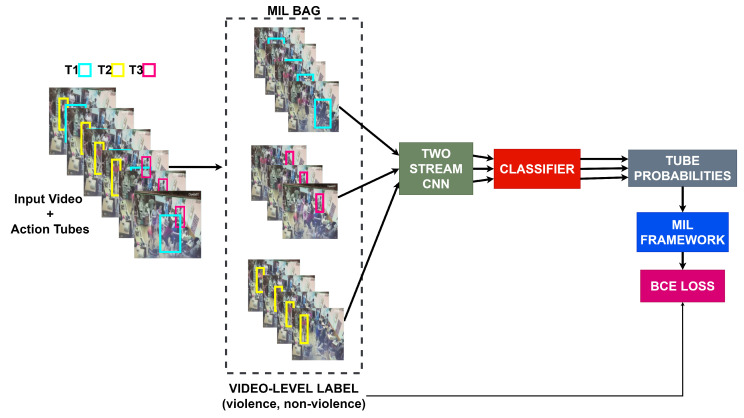
An overview of our approach to training a spatiotemporal violence detector in a weakly supervised manner using an MIL framework. Each bag consists of all tubes (T1,T2 and T3) generated in previous steps. The two-stream model has the same architecture as that presented in Section 3.2. The classifier outputs are passed through a Sigmoid function to obtain tube probabilities. These probabilities are then aggregated by the MIL framework and compared to the bag-level labels during training.

**Figure 8 sensors-22-04502-f008:**
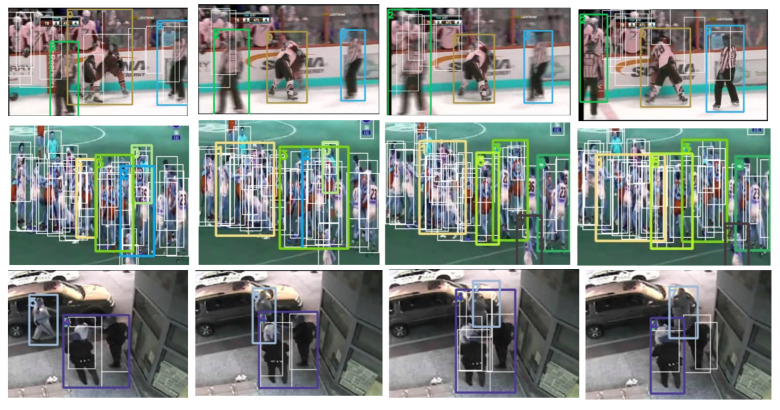
Examples where our proposal generates more than one action tube in a video. Person detection is illustrated with white boxes and action tubes of a different color. The first row shows a video with three action tubes, the second row presents a video with five action tubes, and the third row presents two action tubes for the video.

**Figure 9 sensors-22-04502-f009:**
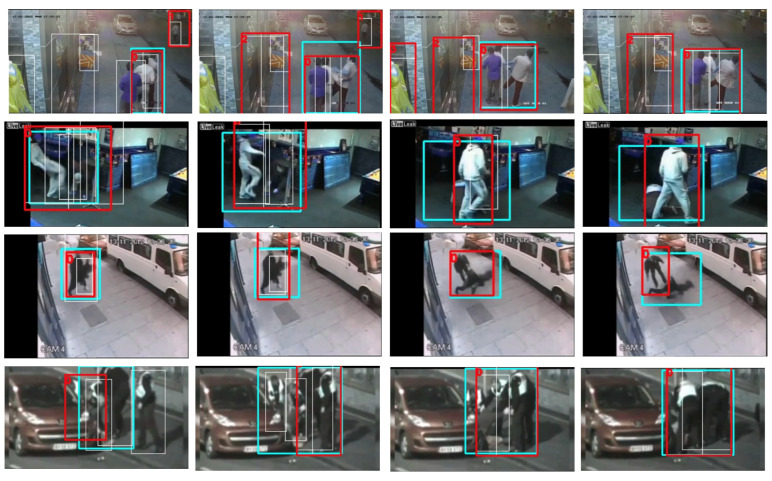
Video samples from the *UCFCrime2Local* dataset with person detections (white boxes), ground truth (light blue boxes), and action tubes (red boxes).

**Figure 10 sensors-22-04502-f010:**
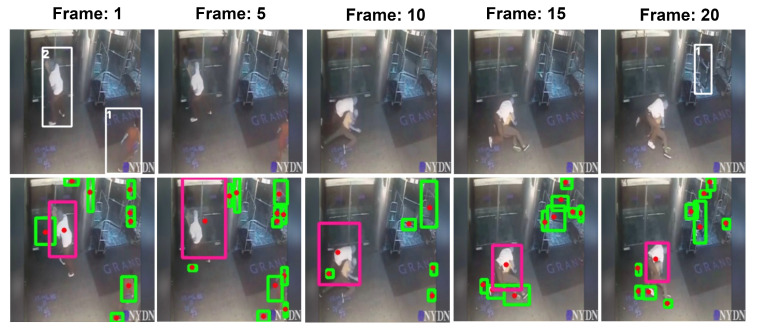
A sequence of frames of a violent video from the RWF-2000 dataset with person detections (white boxes) and final motion regions (pink boxes). The person detector fails in most frames (first row), which produces an action tube that cannot be extracted. In this case, we use the motion regions (second row) to build the action tube.

**Figure 11 sensors-22-04502-f011:**
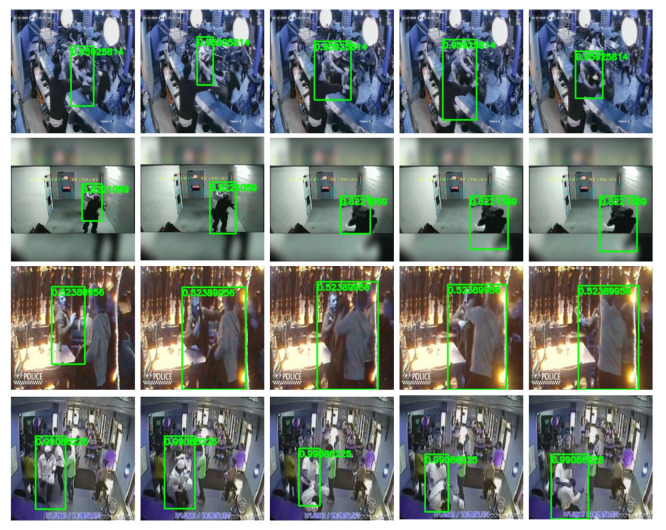
Sample violence localizations for some surveillance videos where each row represents a different video. Green bounding boxes enclose the actors of the violent action, and they are labeled with the violence score predicted by the proposed model.

**Figure 12 sensors-22-04502-f012:**
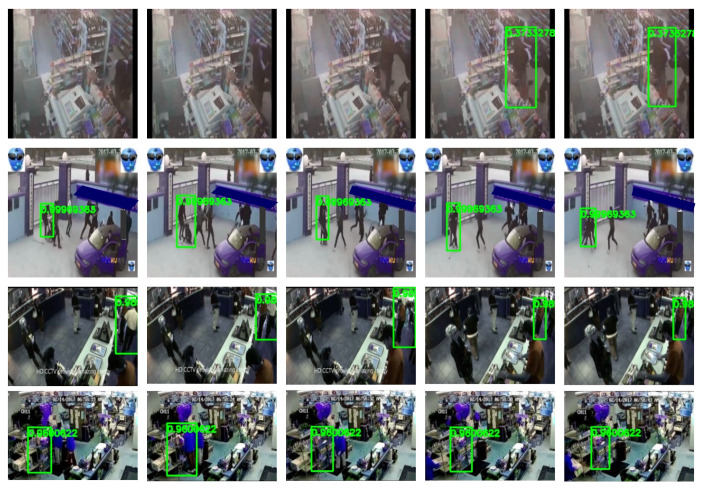
Qualitative examples of failure cases of the proposed model. The action tube and its score are presented in green.

**Table 1 sensors-22-04502-t001:** Analysis of the literature of recent years with respect to spatiotemporal violence detection. Methods with a check detect violent actions both spatially and temporally. Methods with a hyphen only perform temporal detection.

Method	Year	Spatiotemporal Detection
Zhou et al. [7]	2018	-
Deniz et al. [46]	2014	-
Zhang et al. [47]	2016	-
Zhang et al. [42]	2016	√
Bilinski and Bremond [48]	2016	-
Ribeiro et al. [43]	2016	√
Cai et al. [6]	2018	-
Yu et al. [31]	2019	-
Yu et al. [33]	2018	-
Deb et al. [28]	2018	-
Mabrouk and Zagrouba [49]	2017	-
Souza and Pedrini [50]	2017	-
Mohammadi et al. [51]	2017	-
Mahmoodi and Salajeghe [52]	2019	-
Vashistha et al. [53]	2018	-
Febin et al. [54]	2019	-
Ditsanthia et al. [36]	2018	-
Sudhakaran and Lanz [38]	2017	-
Xia et al. [8]	2018	-
Serrano et al. [55]	2018	-
Xu et al. [44]	2018	√
Song et al. [11]	2019	-
Choqueluque-Roman and Cámara-Chávez [45]	2020	√
Cheng et al. [17]	2021	-

**Table 2 sensors-22-04502-t002:** Comparison of the proposed method and other previous methods on the *Hockey Fight* dataset.

Type	Method	Accuracy (%)	AUC
Hand-Crafted Features	Deniz et al. [46]	90.10	0.9480
	Zhou et al. [7]	95.00	0.9798
	Zhang et al. [47]	96.80	0.9808
	Bilinski and Bremond [48]	93.70	-
	Cai et al. [6]	95.80	0.9899
	Yu et al. [31]	95.00	-
	Deb et al. [28]	98.20	-
	Mabrouk and Zagrouba [49]	88.60	0.9323
	Souza and Pedrini [50]	92.79	-
	Mohammadi et al. [51]	81.25	-
	Mahmoodi and Salajeghe [52]	89.30	0.9518
	Vashistha et al. [53]	89.10	-
	Febin et al. [54]	96.50	-
Deep-Learning-Based	Ditsanthia et al. [36]	83.19	-
	Sudhakaran and Lanz [38]	97.10	-
	Xia et al. [8]	95.90	-
	Serrano et al. [55]	94.60	-
	Song et al. [11]	99.62	-
	Choqueluque-Roman and Cámara-Chávez [45]	96.40	-
	Cheng et al. [17]	98.00	-
	Our method	97.30	0.9938

**Table 3 sensors-22-04502-t003:** Comparison of the proposed method and other previous methods on the RWF-2000 dataset.

Method	Accuracy (%)	AUC
Sudhakaran and Lanz [38]	77.00	-
Tran et al. [88]	82.75	-
Carreira and Zisserman [75]	81.50	-
Cheng et al. [17]	87.25	-
Islam et al. [41]	89.75	-
Our method	88.71	0.9149

**Table 4 sensors-22-04502-t004:** Comparison of the proposed method and other previous methods on the *RLVS* dataset.

Method	Accuracy (%)	AUC
Soliman et al. [78]	94.50	-
Moaaz and Mohamed [89]	92.00	-
Jain and Vishwakarma [90]	86.79	-
Our method	92.88	0.9130

**Table 5 sensors-22-04502-t005:** Localization performance on UCFCrime2Local dataset.

Method	Localization Error (%)
Choqueluque-Roman and Cámara-Chávez [45]	35.35
Our method	31.92

**Table 6 sensors-22-04502-t006:** Ablation study of different variants of two-stream models with different fusion strategies. We report the classification accuracy on the RWF-2000 validation set.

	Fusion Strategy	Action Tubes	Accuracy
Two-Stream (RGB + OF)	Late Fusion	No	0.815
Two-Stream (RGB + DI)	Late Fusion	No	0.865
Two-Stream (I3D + 2D ResNet)	Concatenation	Yes	0.866
Two-Stream (I3D + 2D ResNet)	CFAM	Yes	**0.887**
Two-Stream (I3D + 2D ResNet)	CFAM	No	0.83

**Table 7 sensors-22-04502-t007:** Average number of action tubes per video for each dataset extracted with the proposed method.

Dataset	Average Num. Tubes
Hockey Fight [30]	3.14
RWF-2000 [17]	2.61
RLVD [78]	2.64
UCFCrime2Local [79]	3.17

**Table 8 sensors-22-04502-t008:** Ablation study varying the number of action tubes sampled per video. We report the classification accuracy on the RWF-2000 validation set.

Number of Action Tubes	Sampling	Accuracy
1	-	0.8163
2	-	0.8673
3	-	0.8699
4	random	0.8827
4	motion score	0.8871

**Table 9 sensors-22-04502-t009:** Ablation study varying the number of frames per tube. We report the classification accuracy on the RWF-2000 validation set.

Clip Length	Accuracy
8	0.8163
16	0.8827
32	0.8417

**Table 10 sensors-22-04502-t010:** Inference time for each stage on the proposed method.

Task	Time	N° Frames
Person detection	0.0331 s	1
Tube Generation	0.0061 s	16
Tube Classification	0.1992 s	64

## Data Availability

Not applicable.

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
