# Peer review of "Weakly Supervised Violence Detection in Surveillance Video"

_sensors, 2022, doi:10.3390/s22124502_

Round 1

Reviewer 1 Report

The paper proposes a weakly deep supervised approach with the ability to localize spatially and temporally violent actions. It uses pretrained person detectors and an online linking algorithm to build bounding boxes that could contain a violent action. It adapts the Fast-RCNN architecture into the temporal domain for the problem. Experiments are performed on multiple datasets, and results demonstrate that the proposed technique consistently achieves results very compare with the state-of-the-art result in supervised violence detection.

The problem statement that the paper is trying to solve is important and useful. The introduction establishes the severity of the problem of violence detection and there are plentiful references included about violence detection using hand-crafted features, neural networks, and then spatio-temporal techniques. The technical procedure is explained well and the design decisions is well justified. Overall, the work is novel and I recommend acceptance.

Reviewer 2 Report

Theoretical significance is too small.

The experimental comparison of the current version is insufficient.

Reviewer 3 Report

The submission describes a weakly-supervised method to detect spatially and temporarily violent actions in surveillance video using only video level labels. A backbone Fast-RCNN Network is extended and improved. The presented results show the proposed method's improved performance compared to comparable approaches. The overall quality of the paper is in line with the expectation. Following are some of the comments which should be addressed in the revised version of this paper:

  1. In lines 5-6, the author indicates that "most of the previous works focus only on classifying short-clips without performing spatial localization." I do not entirely agree with it, and perhaps some related and recent works may be discussed to improve the quality of literature.
  2. In the Experiments section, the authors should detail what's Accuracy means in violence detection. Besides, there are many evaluation metrics for violence detection in surveillance video, such as AUC, EER, etc. However, the paper only uses Accuracy to evaluate the proposed method; why?
  3. From the view of the application, performance is essential for violence detection. I recommend proposing a small section of time evaluation compared to the other comparable approaches introduced.
  4. Although this paper is well written, there are still some grammar errors in the current version. I suggest the authors carefully proofread this paper and correct all the typos in the revision.

Round 2

Reviewer 2 Report

Authors have greatly improved their manuscript according to the reviewers’ comments.

This paper should be revised the below concerns before it is considered for acceptance.

1.       The authors have to specify the new contributions in this work over the previous works.

2.       Regarding the spatio-temporal detection, please introduce the relevant reference: Integrating object proposal with attention networks for video saliency detection. Inf. Sci. 576: 819-830,2021

3.       Some Figs (e.g. Fig 6) are not clear enough.

4.       Finally, presentation should be further checked throughout the text by a native English speaker.

Reviewer 3 Report

The authors have addressed my concerns.

Author Response

We thank the reviewer for his suggestions and comments.